# The Prognostic Role of Cytogenetics Analysis in Philadelphia Negative Myeloproliferative Neoplasms

**DOI:** 10.3390/medicina57080813

**Published:** 2021-08-09

**Authors:** Giuseppe Lanzarone, Matteo Olivi

**Affiliations:** Unit of Hematology, Department of Oncology, University of Torino, 10126 Turin, Italy; molivi@cittadellasalute.to.it

**Keywords:** myeloproliferative neoplasms, cytogenetics, prognosis

## Abstract

Myeloproliferative neoplasms (MPNs) are clonal stem cell disorders characterized collectively by clonal proliferation of myeloid cells with variable morphologic maturity and hematopoietic efficiency. Although the natural history of these neoplasms can be measured sometimes in decades more than years, the cytogenetics analysis can offer useful information regarding the prognosis. Cytogenetics has a well-established prognostic role in acute leukemias and in myelodysplastic syndromes, where it drives the clinical decisions. NGS techniques can find adverse mutations with clear prognostic value and are currently included in the prognostic evaluation of MPNs in scores such as MIPSS, GIPSS, MIPSS-PV, and MIPSS-ET. We suggest that cytogenetics (considering its availability and relative cost) has a role regarding prognostic and therapeutic decisions.

## 1. Introduction

Myeloproliferative neoplasms (MPNs) are clonal stem cell disorders characterized collectively by clonal proliferation of myeloid cells with variable morphologic maturity and hematopoietic efficiency.

While there are several nosological entities in the Philadelphia negative MPNs classification [1] (Table 1), in this review we will focus on the most prevalent ones: polycythemia vera, essential thrombocythemia and myelofibrosis, while grouping together the other Ph negative MPNs.

Polycythemia vera, essential thrombocythemia and myelofibrosis share three common driver mutations: JAK2 V617F, CAL-R, and MPL are present in both myelofibrosis and essential thrombocythemia, while polycythemia vera harbors JAK2 V617F or mutations in the exon 12 of JAK2. These mutations often constitutionally activate the JAK/STAT signaling pathway [2,3,4]; while other MPNs have different driver mutations: chronic neutrophilic leukemia harbors a mutation of the CSF3 receptor (CSF3R), which locates on chromosome 1p34.3 and encodes the transmembrane receptor for granulocyte colony stimulating factor (G-CSF; CSF3) [5]; chronic eosinophilic leukemia, not otherwise specified (NOS), is defined, among other criteria, as not being a myeloid/lymphoid neoplasm associated with eosinophilia and rearrangement of PDGFRA, PDGFRB, or FGFR1, or with PCM1-JAK2.

Although the natural history of these neoplasms can be measured sometimes in decades more than years, the cytogenetics analysis can offer useful information regarding the prognosis. Cytogenetics is performed conventionally with a karyotype obtained through G-banding of metaphase nuclei from a cell culture; this is usually done on bone marrow, but in some cases, it can be obtained from peripheral blood. Another technique is fluorescence in situ hybridization (FISH) where labeled DNA probes, and, more recently, protein nucleic acid (PNA) probes, can now be hybridized to the chromosome target, detected, and analyzed using fluorescence microscopy [6].

Altered DNA methylation, associated with both age and mutations, also causes DNA breakage, leading to gene deletions (del5q, del7q, and del17p) and duplications (8q and MYC). These are as important prognostically with respect to leukemic transformation as are acquired point mutations [7]. Telomere shortening occurs in myeloproliferative neoplasms, but whether it is associated with aneuploidy is unknown [8].

In this review we will explore the available literature on the prognostic role of cytogenetics in myeloproliferative neoplasms.

## 2. Primary Myelofibrosis

Primary myelofibrosis (PMF) is a clonal Philadelphia chromosome-negative (Ph-) myeloproliferative neoplasm (MPN) characterized by anemia, hepatosplenomegaly, constitutional symptoms (i.e., fatigue, night sweats, fever), cachexia, bone pain, splenic infarct, pruritus, thrombosis and bleeding, and increased risk of leukemic transformation (LT) [9,10]. It can be classified in early/prefibrotic PMF and overt PMF [1]; the former has a less aggressive clinical course but can transform in overt PMF with the progression of the disease. Unlike other Philadelphia negative MPNs, especially Essential Thrombocythemia (ET) and Polycythemia Vera (PV), PMF is associated with a significant decrease in life expectancy in comparison with age- and sex-matched controls from the general population, with a median overall survival (OS) of 6.5 years [11].

In 2008, Cervantes et al. series identified different clinical factors implicated in PMF prognosis, namely anemia (hemoglobin [Hgb] <10 g/dL), leukocytosis, age, constitutional symptoms, circulating blasts, abnormal karyotype, percentage of blood myeloid precursors, thrombocytopenia, number of circulating progenitors, monocyte count, and male sex [12]. The rising interest in PMF prognostic modelling has led to the publication of different scores, aiming to refine the risk-stratification of this population.

In 2009, Cervantes et al. firstly developed the International Prognostic Scoring System (IPSS) based on characteristics of 1054 PMF patients. A list of five parameters (age >65 years, constitutional symptoms, Hgb <10 g/dL, leukocyte count >25 × 10^9^/L, and circulating blasts >1%) associated with shorter survival were able to stratify PMF patients in four risk categories (low-, intermediate 1-, intermediate 2-, high-risk) with significantly different OS (135 months, 95 months, 48 months, and 27 months, respectively) [13].

The IWG-MRT (International Working Group for Myelofibrosis Research and Treatment) then published a dynamic prognostic model, the Dynamic IPSS (DIPSS), which made possible to evaluate the IPSS prognostic factors at later time points than at diagnosis. Multivariable Cox regression analysis performed including IPSS factors as time-dependent covariates showed that acquisition of severe anemia during the disease course affected survival more severely than the others. Thus, in the DIPSS, Hb <10 g/dL was assigned 2 points, whereas the others continued to be assigned 1 point each [14].

Although not considered in above mentioned models, distinct studies clearly demonstrated that cytogenetics abnormalities affect survival of patients with myelofibrosis in an IPSS- and DIPSS-independent manner [15,16], with 2-years mortality ranging from 60% to >80% in presence of adverse karyotype alterations or monosomal karyotype (MK) and inv(3)/i(17q), respectively [17].

Conventional cytogenetics studies revealed the absence of a specific abnormality in PMF. However, recurrent non-random karyotypic alterations are detectable in almost one-third of PMF patients [18]. Hussein et al. previously published an interesting review summarizing more common findings in PMF cytogenetic analysis [19]. Considering literature publications [20,21,22,23,24,25,26,27], most frequent abnormalities were del(20q) (22% [20], 34% [21], and 36% [22] of abnormals), del(13q) (14% [20], 24% [21], and 25% [22] of abnormals) and chromosome 1 abnormalities (19–28% [20,21,22] of abnormals). Additional abnormalities commonly identified were +8 and +9. Less frequently, abnormalities of chromosome 3, −5⁄del(5q), −7/del(7q), del(12p), and +21 were observed [19].

The identification of common recurrent anomalies in cytogenetic analysis made necessary to attribute a prognostic significance to these findings. Three different studies, comprising 202 [28], 200 [15], and 131 [29] patients with PMF diagnosis outlined the favorable prognosis associated with isolated 13q− and isolated 20q−. In addition, Hussein et al. and Tam et al. identified isolated +9 as a favorable prognostic molecular marker [15,29]. Unfavorable abnormalities identified in these and other related studies consist in complex karyotype (>3 abnormalities), isolated +8 and an abnormal karyotype with abnormalities of chromosomes 5, 7, 17, or 12p− [15,28,29,30]. These observations allowed the Mayo Clinic group to publish two subsequent studies aiming to refine risk stratification in this subpopulation [15,16]. Specifically, in 2010, Hussein et al. research considering 200 PMF patients outlined an IPSS-independent cytogenetic risk categorization consisting in 4 clearly distinct groups: “favorable” (−9, 20q−, 13q−), “normal”, “other abnormalities” and “unfavorable” (+8 or complex karyotype) with a median OS of 113, 80, 46, and 34 months, respectively [15]. Afterwards, a study with 433 patients of the same research group, comprising the 200 of the previous one, led to the elaboration of a two-tailored risk stratification model: “favorable” group included isolated 13q−, isolated 20q−, isolated + 9, isolated chromosome 1 translocation/duplication, ‘other sole abnormalities of indeterminate-risk’ or ‘two abnormalities excluding an unfavorable type’ and normal karyotype; “unfavorable risk” included isolated +8, isolated −7/7q−, ‘other high-risk sole abnormalities [i(17q), −5/5q−, 12p−, 11q23 rearrangement or inv(3)]’ or ‘two abnormalities including group unfavorable type’, and complex karyotype (>3 abnormalities) [16]. These studies were crucial because they were able to identify a risk categorization capable to discriminate significant differences in overall survival (OS) and leukemia-free survival (LFS). Moreover, they clearly revealed that cytogenetics has prognostic relevance irrespective of previously elaborated scores.

These data, together with the discovery of the detrimental prognosis associated with thrombocytopenia and red blood cell transfusions [16,31], led to the development of the DIPSS-plus score [32]. In this case, the DIPSS low-risk, intermediate-1-risk, intermediate-2 risk, and high-risk categories were assigned 0, 1, 2, and 3 points respectively, whereas unfavorable karyotype (complex or a single or 2 abnormalities including +8, −7/7q−, i(17)q, −5/5q−, 12p−, inv(3), or 11q23 rearrangements), a platelet count <100 × 10^9^/L, and RBC transfusion dependence were assigned 1 point each.

A more recent work of Tefferi et al. with 1002 patients in 2018 further revised this stratification, identifying a three-tiered risk model: ‘very high risk (VHR)’: single/multiple abnormalities of −7, i(17q), inv(3)/3q21, 12p−12p11.2, 11q−/11q23, or other autosomal trisomies not including +8/+9 (e.g., +21, +19); ‘favorable’: normal karyotype or isolated abnormalities of 13q−, +9, 20q−, chromosome translocation/duplication or sex chromosome abnormality including -Y; ‘unfavorable’: all other abnormalities. The current model was able to predict survival effectively and its impact was independent of IPSS and DIPSS and concomitant “driver” and “not driver” mutations. Furthermore, it maintained the ability to predict leukemic transformation (LT), with HRs of 4.4 and 2.0 for VHR and for unfavorable groups, respectively [33].

All these considerations highlight the crucial role of cytogenetics analysis in primary myelofibrosis. Its feasibility, the relative cost, and its ability in predicting OS and LFS, regardless of clinical parameters historically considered, made it a cornerstone of modern prognostic scores. MIPSS (mutation-enhanced international prognostic score system) score firstly integrated karyotypic and clinical variables with mutational status of CALR, ASXL1, EZH2, SRSF2, and IDH1/2 [34] (plus U2AF1 in MIPSS plus version 2.0 score [35]). Given that karyotype conserved MIPSS-independent prognostic value for both OS and LFS, a newer GIPSS (Genetically Inspired Prognostic Scoring System) score based exclusively on age, karyotype, and mutations was developed, aiming to completely replace clinical variables with genetic markers, for prediction of survival in PMF [36]. According to that, recently published guidelines for the management and treatment of primary myelofibrosis integrate karyotype and mutational tests as essential elements in guiding routine clinical decisions [10].

## 3. Secondary Myelofibrosis

About 10% to 15% of PV and 1% to 5% of ET evolve to myelofibrosis after several years, resulting in a “secondary” myelofibrosis known as “post polycythemia vera myelofibrosis” (PPV-MF) or “post essential thrombocythemia myelofibrosis” (PET-MF) [9]. The clinical and hematologic characteristics of PPV-MF/PET-MF are similar to those of primary myelofibrosis, associating anemia with teardrop poikilocytosis, marked splenomegaly, leucoerythroblastic blood picture, increased circulating CD34+ progenitors, and bone-marrow fibrosis on histologic examination.

In a series of 30 patients with PPV-MF, when the hematologic status changed to a myelofibrotic picture (PPV-MF), the proportion of normal karyotypes fell to 10% and 27 of 30 patients showing an abnormal cytogenetic clone: in some patients, the researchers found cytogenetic abnormalities classically observed in PV, such as del(20)(q12) in three cases, and del(7q), del(11q), as well as i(17) (q10) in single cases. Only one patient showed a more complex karyotype. Most patients had unbalanced translocations, with or without additional cytogenetic changes, leading to total or partial trisomy 1q: implicated chromosomes were chromosome 14 (2 cases), chromosome 9 (2 cases), chromosome 15 (2 cases), chromosome 7 (2 cases), 13p, 19q, 16p, 1q, 6p, and chromosome Y, one case each. Other partial structural changes, such as i(1)(q10) (one patient) leading to tetrasomy 1q and dup(1q) (patients 10, 11, 20, and 27) also resulted in partial trisomy 1q [37].

In a series of 66 patient aged <60 years, restricting the analysis to those with available cytogenetics information (31 patient), the presence of unfavorable cytogenetic abnormalities (abnormalities other than isolated 13q− and 20q−) became the only independent prognostic factor for inferior survival [38].

Although the prognostic role of cytogenetics is well established in primary myelofibrosis, its role is much less clear in the PPV-MF and PET-MF: the MYSEC-PM (Myelofibrosis Secondary to PV and ET-Prognostic Model) score, developed in a cohort of 262 PPV-MF and PET-MF patients [39], does not include cytogenetics data in the prognostic assessment. The score was later validated in a cohort of patients treated with ruxolitinib, maintaining its prognostic significance [40].

## 4. Polycythemia Vera

According to 2016 WHO classification of myeloid malignancies, polycythemia vera (PV) is one of the Philadelphia chromosome-negative myeloproliferative neoplasms (MPNs), typically characterized by increased red blood cell production. Nonetheless, red cell mass increase with higher level of hemoglobin (>16.5 g/dL in men and >16 g/dL in women) or hematocrit (>49% in men and >48% in women) is not sufficient for diagnosis, and bone marrow evaluation together with the presence of JAK2 mutation and low EPO serum level are now required [1]. Median overall survival is approximately 14 years [41]; the corresponding value for patients younger than 60 was 24 years.

Historically, advanced age (>60 years) and history of prior thrombosis helped clinicians in identifying high-risk PV patients, due to their association with the risk of thrombotic events, that still constitute the leading cause of preventable death in PV [42]. Unlike PMF, as discussed above, cytogenetics analysis and abnormalities, the focus of our review, do not have a clear prognostic relevance in PV patients. Few studies have focused on this subject and its clinical significance is still uncertain.

In 2008, Gangat et al. failed to demonstrate a statistically significant correlation between cytogenetics abnormalities at diagnosis and parameters like JAK2 V617F allele burden, thrombosis, hemorrhage, leukemic/fibrotic transformation, or survival [43]. Similarly, the work of Swolin et al. highlighted a similar lack of association with the transformation in myelofibrosis or acute myeloid leukemia [44].

More recently, an international study with a greater number of patients and a more adequate follow-up time including 1545 patients with PV discovered factors influencing survival. Among those taken into consideration, age, leukocytosis, history of venous thrombosis, and abnormal karyotype were identified as independent risk variables predicting detrimental survival. In the same work, abnormal karyotype (HR 3.9; 95% CI) emerged also as one of the independent risk factors for leukemia-free survival (LFS), together with older age and leukocytosis [45]. A few years later, in 2018, the same research group confirmed previously mentioned results considering a cohort of 196 patients with a median follow up of 84 months. Cox proportional hazard regression model employed for multivariate analysis for overall survival (OS) and leukemia-free survival (LFS) revealed the adverse effect of abnormal karyotype (HR 1.9 and 15.8, respectively) [46].

At least 14–20% of patients with PV carries out at least one karyotype abnormality at time of initial diagnosis. Among them, in several independent studies del(20q), +8, +9 and +1q were the most commonly reported [18,43,44,47,48]. Nonetheless, the prognostic impact of individual cytogenetic abnormalities was not further classified, and none of the above-mentioned alterations have been shown conclusively to have prognostic value [47,49]. To date, Tang et al. study is the sole trying to answer this question [50]: considering retrospective data of 422 PV patients, they confirmed del(20q), +8, +9 and +1q as the most frequent cytogenetic lesions at diagnosis. A more exhaustive analysis conducted at different stages of the disease revealed distinct abnormalities in different phases with del(20q), +8, +9 having a higher incidence in polycythaemic phase and complex karyotype (including more frequently −5/del(5q), −7/del(7q), −17/del(17p)/i(17q), and −18) characterizing the accelerate/blastic phase (defined as PV with ≥10% blasts or ≥20% blasts in PB or BM or both, respectively). Moreover, the study clearly demonstrated that more advanced stages correlate with a higher frequency of abnormal karyotype (20% in PP, 90% in AP/BP) [50]. Results concerning the survival impact of specific chromosomal abnormalities in patients in polycythaemic phase, led Tang et al. to stratify this population in three distinct groups with significant differences in OS: a “low-risk” group with a normal karyotype, sole +8, sole +9, and other single abnormalities with a median OS of 169 months; an “intermediate-risk” group with sole del(20q) and double abnormalities (including +1q) (median OS of 86 months); a “high-risk” group including complex karyotype, having a shorter median OS of 9 months [50].

Recent research has focused on integrating genetic, cytogenetics, clinical, and laboratory data in a new prognostic model for ET and PV (MIPSS-ET and MIPSS-PV): a total of 906 molecularly annotated patients (416 from Mayo and 490 from Florence), including 502 ET and 404 PV cases, were included in the current study. Considering PV subpopulation, the Mayo/Florence cohorts included 146/258 (median age 63/58 years, 52/44% females) patients. The presence of adverse mutations independently affected survival in PV, in both the Mayo and Florence’s patient cohorts; HRs (95% CI) were 3.4 (1.2–8.6) for Mayo/PV and 9.8 (2.2–44.3) for Florence PV. Mutation-independent risk factors for OS in PV/Mayo included age >67 years (5.9, 3.2–11.2), leukocyte count ≥15 × 10^9^/L (3.5, 1.9–6.3), and abnormal karyotype (2.5, 1.2–4.7); in PV/Florence, age >67 years (3.2, 1.7–6.4) and leukocyte count ≥11 × 10^9^/L (2.1, 1.1–4.0) [51].

Given the data of the studies above, and the need of further confirmations of its results, prognostic implications of an abnormal karyotype, both on OS and LFS, has emerged in the literature [46,50,51], and suggests the practical importance of obtaining cytogenetic information at time of diagnosis in PV, and possibly during its clinical course.

## 5. Essential Thrombocytopenia

Essential thrombocythemia (ET) is a myeloproliferative neoplasm (MPN) characterized by persistent elevated platelet count (>450 × 10^9^/L) in peripheral blood, bone marrow megakaryocytic proliferation with large and mature morphology, and presence of JAK2V617F/CALR/MPL clonal mutation [1]. Usually, clinical course of the disease is benign with a life expectancy similar to that of healthy individuals (approximately 20 years) [41].

However, an increased risk of thrombosis and bleeding, and transformation to an aggressive myeloid disorder, such as myelofibrosis or acute leukemia, can happen in the natural history of this MPN. Molecular studies in myeloproliferative diseases in recent years has led to a significant step forward in their pathogenetic comprehension. Concerning ET, almost 55% of patients harbor the JAK2 V617F mutation, while CALR and MPL are present in nearly 15–24% and 4% of patients, respectively [52,53,54]. JAK2 V617F associates with older age, increased thrombotic risk, higher hemoglobin level, leukocytosis, and lower platelet count [2,55]. CALR mutation correlates with younger age, male sex, higher platelet count, lower hemoglobin level, lower leukocyte count, and lower incidence of thrombotic events, with type 2 vs. type 1 CALR mutations associated with higher platelet count [56,57]. MPL mutations have been inconsistently associated with older age, female gender, lower hemoglobin level and higher platelet count, while possible association with inferior myelofibrosis- free survival has been reported in ET [54,58,59]. 10–15% of patients do not harbor one of the known mutations and are called “triple negative”; in some cases, with additional investigations, a gain-of-function mutation in MPL or JAK2, either somatic or hereditary, can be found [60].

On the other hand, cytogenetic analysis in ET helped little in understanding disease pathogenesis. This is primarily due to the rarity and the heterogeneity of karyotype anomalies. Rate of their frequency is between 1 and 7%, including both structural and numerical changes [42,61,62,63,64,65,66], usually distributed on every chromosome with no specific genetic marker. Most frequent chromosomal abnormalities reported were del(20q), del(13q), del(12p), trisomy 8, trisomy 9, partial duplication of 1q, balanced translocations involving 8p11 and gains in 9p, as also observed in other myeloid malignancies [67,68].

Concerning prognostic impact of cytogenetics in ET, few studies often with only a small number of patients have been published to date, making difficult to attribute a real relevance to it. Gangat et al. [62] reported an association between abnormal karyotype and palpable splenomegaly, current tobacco use, venous thrombosis, and anemia but no effect on survival. Sever et al. [66], similarly, concluded that there is no difference in survival of patients with cytogenetic abnormalities and those with normal karyotype at diagnosis. Hsiao et al. [63] in line with previously mentioned reports and others, failed to demonstrate a relation between chromosomal aberrations at presentation and inferior survival or disease transformation. However, an association between an increased risk of transformation and detrimental survival was found with de novo appearance of cytogenetic abnormalities in ET, attributing an importance to a sequential cytogenetic follow-up during the disease’s course, contrarily to Sever et al. conclusions [66].

Taking into consideration the above-cited publication [51] and focusing on ET cases, a total of 502 patients were included in the study (270 in Mayo Clinic cohort and 232 in Florence one with a median age of 57 and 54 years, respectively). As well as in PV subcohort, the presence of adverse mutations independently affected survival, in both the Mayo and Florence’s patients (HRs 95% CI 2.6 (1.3–4.7) for Mayo/ET and 2.0 (1.1–3.6) for Florence ET). Similar analysis on mutation-independent risk factors for OS were conducted individuating for the Mayo cohort, age >60 years (5.4, 3.3–9.1) and male gender (1.7, 1.1–2.6), and for the Florence cohort, age >60 years (12.4, 6.3–24.4), male gender (1.6, 1.0–2.7) and leukocyte count ≥11 × 10^9^/L; karyotype in the Mayo ET cohort was borderline significant (*p* = 0.06) [51].

In summary, ET cytogenetics seems to be of secondary interest both because of the relative rarity of aberrations and the lack of a real predictive prognostic significance. Analogously, the importance of repeating cytogenetics analysis during clinical follow-up when transformation is suspected needs to be confirmed in further studies.

## 6. Other Ph-Myeloproliferative Neoplasms

The 2016 WHO classification of Philadelphia chromosome negative myeloproliferative neoplasms includes less frequent pathologic entities other than ET, PV, or PMF that we concisely discuss in this paragraph [1].

### 6.1. Chronic Neutrophilic Leukemia (CNL)

Chronic neutrophilic leukemia (CNL) is a rare disease characterized by sustained leukocytosis (≥25 × 10^9^/L) with clonal proliferation of mature neutrophilic granulocytes in blood and bone marrow [1]. Unlike chronic myeloid leukemia (CML), the abnormal sustained proliferation involves only the neutrophilic lineage [69]. In 2013, discovery of the CSF3R mutation gave essential information about pathogenesis of the disease: CSF3R locates on chromosome 1p34.3 and encodes the transmembrane receptor for G-CSF/CSF3; it plays an important role in proliferation and differentiation of granulocytes [5]. In terms of cytogenetics, no specific genetic markers have been identified in CNL, with many patients displaying normal karyotype. Reilly’s review series estimated an abnormal karyotype in 37% of CNL cases [70]. Most frequent anomalies include +8 [71], +21 [72], deletion 11q [73], and deletion 20q [74]; the latter being identified as the most common [71]. None of the above mentioned, however, resulted specific for CNL or able to predict survival.

### 6.2. Chronic Eosinophilic Leukemia, NOS (CEL, NOS)

Chronic eosinophilic leukemia is not otherwise specified (NOS) and is also one of the entities listed under the MPN category in the WHO classification. It consists of a distinct disease from myeloid/lymphoid neoplasms associated with eosinophilia and rearrangement of PDGFRA, PDGFRB, or FGFR1, or with PCM1-JAK2. Specifically, the diagnosis of CEL-NOS requires: (a) peripheral eosinophils count of 1.5 × 10^9^/L or more; (b) absence of WHO criteria for BCR-ABL1-positive chronic myeloid leukemia, PV, ET, PMF, CNL, CMML, or atypical CML; (c) absence of rearrangement of PDGFRA, PDGFRB, or FGFR1; no PCM1-JAK2, ETV6-JAK2, or BCR-JAK2 fusion gene; (d) blast count in peripheral blood or bone marrow <20% without diagnostic features of AML; (e) presence clonal cytogenetic or molecular genetic abnormality, or blast cells are ≥2% in the peripheral blood or >5% in the BM [1]. An evidence of clonality clearly emerged in past studies with more than 90% of cases showing an abnormal karyotype. The most frequent abnormalities historically that were identified by conventional cytogenetics are +8, +15, del(5q), del(9), −7, i(17q), or a complex karyotype, all of which are not unique to CEL-NOS and apparently without clear prognostic implications [75].

### 6.3. Mastocytosis

Mastocytosis is a rare disorder that is characterized by heterogeneous clinical manifestations ranging from benign isolated infiltration of the skin, urticaria pigmentosa, to a generalized disease, systemic mastocytosis (SM). Listed among MPNs in 2016 WHO classification, mastocytosis consists in clonal proliferation of mast cells and includes five major subtypes: indolent SM (ISM), smouldering SM (SSM), SM with an associated clonal hematopoietic non-MC disease (SM-AHN), aggressive SM (ASM), and MC leukemia (MCL); the last three defined as “advanced SM” (AdvSM) [76].

Clonality clearly emerged after the discovery that most patients affected shared one of the mutations of the c-KIT gene. The gene, encoding a stem cell factor receptor, permits the growth of mast cells independently from growth factors and sustains their tumoral multiplication [77]. As stated above for other Ph- MPNs, a distinctive recurrent cytogenetic does not seem to be recognizable in mastocytosis. However, past studies revealed that a significant proportion of cases presented clonal cytogenetic aberrations [78,79,80]. Among them were included some that have already been reported in other myeloid neoplasms, such as 5q−, 11q−, 20q−, and +8. Other cases have shown single metaphases with an abnormality characteristic of a myeloid neoplasm, including 7q−, 11q−, and 20q− [80]. More recently, a German group analyzed the incidence and prognostic impact of cytogenetic abnormalities in 109 patients with indolent (*n* = 26) and advanced (*n* = 83) mastocytosis with (*n* = 73; 88%) or without associated hematologic neoplasm (AHN). In the considered population, karyotypic anomalies were found only in SM-AHN patients. When patients were stratified according to their karyotype abnormality into a good-risk group (*n* = 73; normal or favorable karyotype: del(5q); trisomy 8; del(1q); del(12p)) and a poor-risk group (*n* = 10; complex karyotype (defined as ≥3 abnormalities); monosomy 7; del(5q)), the second showed a significantly shorter OS [81]. A similar analysis in 348 patients with SM was performed by the Mayo Clinic group. The overall incidence of cytogenetic abnormalities was 15% all SM subtypes considered, once again with a higher prevalence in the SM-AHN subgroup (26%). In this case, an abnormal karyotype was associated with inferior OS only in univariate analysis [82].

### 6.4. Myeloproliferative Neoplasm, Unclassifiable

Myeloproliferative neoplasm, unclassifiable (MPN-U), has clinical, laboratory and morphological features of an MPN but fails to meet the criteria for any of the specific MPN entities [1].

In a study of 10 patients diagnosed with MPN-U based on the histological findings of the bone marrow, six of eight evaluable patients had a JAK2 V617F mutation, and of the 10 patients, +8 was found in two patients and t(6;12) (q12;p13) in one patient [83].

In a series of 23 patients with a del5q 4/23 had a diagnosis of MPN-U, while PMF was the most common [84].

Among 13 patients with MPNs, two had MPN-U and progressed to secondary myelofibrosis; the single nucleotide polymorphism array (SNP-A)-based karyotyping showed abnormal results, whereas conventional metaphase cytogenetics (MC) analysis showed normal results at diagnosis and during follow-up. The alterations were loss of 6q, copy neutral loss of heterozygosity (CN-LOH) of 9p, 14q, 19p [85].

Chamseddine et al. reported a case of a MPN-U with a BCR-JAK2 translocation t(9;22) (p24;q11); the patient had an aggressive disease course, and the authors speculate that it could be caused by the translocation itself; they suggest allogeneic hematopoietic stem cell transplantation (allo-HSCT) as the most effective therapy [86].

The cytogenetics data in these rare neoplasms are scarce and reflect abnormalities found in other myeloid neoplasms; their prognostic value in this setting thus cannot be clearly defined.

## 7. Conclusions

Cytogenetics has a well-established prognostic role in acute leukemia (both myeloid and lymphoid [87]) and in myelodysplastic syndromes [88], where it drives the clinical approach to decide the best therapy for the patient (i.e., indication to allo-HSCT or not).

The most common cytogenetics alterations are summarized in Table 2. The alterations are not specific of a single disease but are shared across myeloid malignancies, as reported by Mukherjee et al. [89].

In the setting of PMF, the prognostic role of cytogenetics is currently used to guide therapeutic options in scores like DIPSS-plus and MIPSS70-plus 2.0, while in other MPNs it is much less defined and does not inform therapeutic options routinely.

Cytogenetics has some limitations and drawbacks, in particular, having to perform the analysis on a fresh sample of adequate cellularity; the cell culture must be checked at 24 or 48 h and cannot yield enough metaphase nuclei for the analysis; and the G-banded karyotype should be performed by an experienced operator.

Next generation sequencing (NGS) techniques are DNA sequencing which have revolutionized genomic research, permitting the study of millions of small fragments of DNA in parallel, with enormous reduction in time in respect to classical Sanger sequencing [90]. The main drawback is the cost of the required infrastructure and expert personnel; thus it is becoming more common but not yet widespread outside academic research centers.

NGS techniques can find adverse mutations with clear prognostic value and are currently included in the prognostic evaluation of MPNs in scores, such as the MIPSS, GIPSS, MIPSS-PV, and MIPSS-ET [34,35,36,39,51].

DNA alterations are not only in the sequence (as explored with NGS techniques) or in the number/structure of chromosomes (as determined by cytogenetics analysis/FISH), but also in the methylation and gene expression: methylome profiling is different between ET/PV and PMF, and a different profile is associated with TET2 and ASXL1 mutations; the latter having a clear prognostic value [34,91].

However, even in the era of NGS, we suggest that cytogenetics (considering its availability and relative cost) should be performed in every suspected myelofibrosis and could be performed in myeloproliferative neoplasms at diagnosis or at the time of suspected transformation (Table 3). Determining a baseline karyotype helps the clinician determine prognosis in MF, PV, and SM; the prognostic value in ET, MPN-U, CNL, and CEL is unclear, but in these diseases the cytogenetics analysis at diagnosis can give information about clonal evolution and track clonal changes at the time of transformation.

## Figures and Tables

**Table 1 medicina-57-00813-t001:** WHO 2016 classification of myeloid neoplasms and acute leukemia.

WHO 2016 Myeloid Neoplasm and Acute Leukemia Classification
Myeloproliferative neoplasms (MPN)
○ Chronic myeloid leukemia (CML), BCR-ABL1+
○ Chronic neutrophilic leukemia (CNL)
○ Polycythemia vera (PV)
○ Primary myelofibrosis (PMF)
○ PMF, prefibrotic/early stage
○ PMF, overt fibrotic stage
Essential thrombocythemia (ET)
○ Chronic eosinophilic leukemia, not otherwise specified (NOS)
○ MPN, unclassifiable
Mastocytosis

**Table 2 medicina-57-00813-t002:** Common cytogenetics alterations in MPNs.

Diagnosis	Frequent Cytogenetic Alterations	References
PMF	Del(20q) Del(13q), chromosome 1 abnormalities	[20,21,22]
PPV-MF/PET-MF	Chromosome 1 abnormalities Del(20q)	[37,38]
Polycythemia vera	Del(20q) +8 +9	[50]
Essential thrombocythemia	Del(20q) Del(13q) +8 +9	[67,68]
Chronic Neutrophilic Leukemia	Del(20q) Del(11q) +8 +21	[70,71,72,73,74]
Chronic Eosinophilic Leukemia	+8 +15 Del(5q) i(17q) −7	[75]
Mastocytosis	Del(20q) Del(11q) Del(7q)	[80]
Myeloproliferative neoplasm, Unclassifiable	+8 Del(5q)	[83,84]

**Table 3 medicina-57-00813-t003:** Suggested cytogenetics timing.

When to Perform Cytogenetics	Diagnosis	Routine BM Examination, If Performed	Time of Suspected Transformation
Myelofibrosis	Always if feasible	Always if feasible	Always if feasible
Polycythemia vera	Always if feasible	No	Always if feasible
Essential thrombocythemia	Always if feasible	No	Always if feasible
Chronic Neutrophilic Leukemia	Always if feasible	No	Always if feasible
Chronic Eosinophilic Leukemia	Always if feasible	No	Always if feasible
Mastocytosis	Always if feasible	No	Always if feasible
Myeloproliferative neoplasm, Unclassifiable	Always if feasible	No	Always if feasible

## Data Availability

Not applicable.

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
