# Peer review of "The Prognostic Role of Cytogenetics Analysis in Philadelphia Negative Myeloproliferative Neoplasms"

_medicina, 2021, doi:10.3390/medicina57080813_

Round 1
Reviewer 1 Report
The review is well written. Here are my comments:
Line 51: PCM1-JAK2 instead of PCM-JAK21 – please correct
Line 68: Please correct MF to PMF since this abbreviation is used later on
Line 134: Reference number is not formatted as it should be.
Line 153: Reference number is fused with IDH1/2
Line 154: Reference number!!
Lines 322-323: Abbreviations for GCSF and CSF3 are already defined, no need to repeat it
In general:
- Please check the reference number formats (at some lines, the numbers are fused with the preceding word and not formatted as they should be “[XX]”.
- Pre-MF should also be mentioned since it needs to be discriminated from other MPN's – it has prognostic and therapeutic implications.
- Please also define other abbreviations, ie. MIPSS
- It would be helpful for the reader to see a table with the most common cytogenetic aberrations across all mentioned Ph-negative MPN’s and their occurrence in particular entities
Author Response
Line 51 corrected; Line 68 corrected; Line 134 corrected; Line 153-154 corrected; Line 322-323 corrected
MIPSS and GIPSS abbrevation were clarified.
The prefibrotic PMF patients were included in the studies cited, so we added a clarifying line.
Added a table with the most common cytogenetics alterations
Reviewer 2 Report
In this comprehensive review, Lanzarone and Olivi present an interesting and thought-provoking review of the role of cytogenetics in prognostication of Ph- MPNs. The manuscript also highlights the challenges and limitations encountered by the cytogenetics community while attempting to use cytogenetics specifically in rare Ph- MPNs and propose a cytogenetic work up for these diseases. The notions in the manuscript would certainly benefit the field.
- The key concept the authors are trying to put forth is that performing timed cytogenetics might have prognostic role in all Ph- MPNs. While this is compelling and true for many of these diseases, it is unclear whether this would have a prognostic role in some of the Ph- MPNs such as ET, CNL, CEL and MPN-U. The manuscript and the readers would benefit if the author can add some information about how performing cytogenetics using their proposed scheme could improve the prognostic value in these diseases (ET, CNL, CEL and MPN-U).
- A few lines about the potential limitations of cytogenetics could also be discussed. In addition, recent studies have indicated that Ph- MPNs harbor mutations in epigenetic modifiers such as TET2 indicating aberrant epigenetic landscape in these patients. The authors could consider adding a few lines about how DNA methylomes could be used for prognostication in addition to other technologies in the conclusion section.
- Please include expansions for all the abbreviations once in the text. This would help readers from outside the field. Below are some which missed expansions.
Line 88 – IWG-MRT
Line 152 – MIPSS
Line 155 – GIPSS
Line 184 – MYSEC-PM
Line 349 – add SM for Systemic Mastocytosis
- Typos and references
Line 51 – JAK2
Line 120 – respectively [15]
Line 134 – score [32]
Line 236 - 2and to and
Line 327 - +8 [71]
Line 366 - anomalies
Author Response
IWG - MRT clarified; MIPSS clarified, GIPSS clarified, MYSEC-PM clarified; SM added
Typos and reference were fixed
Added clarification to our proposed cytogenetics scheme, to the limitation of cytogenetics
Added a comment about DNA methylome